# Cannabidiol and Cannabigerol Inhibit Cholangiocarcinoma Growth In Vitro via Divergent Cell Death Pathways

**DOI:** 10.3390/biom12060854

**Published:** 2022-06-20

**Authors:** Michael J. Viereckl, Kelsey Krutsinger, Aaron Apawu, Jian Gu, Bryana Cardona, Donovan Barratt, Yuyan Han

**Affiliations:** 1School of Biological Sciences, University of Northern Colorado, Greeley, CO 80639, USA; michael.viereckl@unco.edu (M.J.V.); kelsey.krutsinger@unco.edu (K.K.); card8263@bears.unco.edu (B.C.); 2Department of Chemistry and Biochemistry, University of Northern Colorado, Greeley, CO 80639, USA; aaron.apawu@unco.edu; 3Department of Epidemiology, The University of Texas MD Anderson Cancer Center, Houston, TX 77030, USA; jiangu@mdanderson.org; 4School of Biological Sciences, Iowa State University, Ames, IA 50011, USA; barr5773@iastate.edu

**Keywords:** cholangiocarcinoma, cannabidiol, cannabigerol, cell death

## Abstract

Cholangiocarcinoma (CCA) is a rare and highly lethal disease with few effective treatment options. Cannabinoids, cannabidiol (CBD) and cannabigerol (CBG) are non-psychedelic components extracted from cannabis. These non-psychoactive compounds have shown anti-proliferative potential in other tumor models; however, the efficacy of CBD and CBG in CCA is unknown. Furthermore, two cell death pathways are implicated with CBD resulting in autophagic degeneration and CBG in apoptosis. HuCC-T1 cells, Mz-ChA-1 cells (CCA cell lines) and H69 cells (immortalized cholangiocytes), were treated with CBD and CBG for 24 to 48 h. The influence of these cannabinoids on proliferation was assessed via MTT assay. Apoptosis and cell cycle were evaluated via Annexin-V apoptosis assay and propidium iodide, respectively. The expression of proliferation biomarker Ki-67, apoptosis biomarker BAX, and autophagic flux biomarkers LC3b and LAMP1 were evaluated via immunofluorescence. Cell migration and invasion were evaluated via wound healing assay and trans-well migration invasion assays, respectively. The colony formation was evaluated via colony formation assay. In addition, the expression of autophagy gene LC3b and apoptosis genes BAX, Bcl-2, and cleaved caspase-3 were evaluated via Western blot. CBD and CBG are non-selective anti-proliferative agents yielding similar growth curves in CCA; both cannabinoids are effective, yet CBG is more active at lower doses. Low doses of CBD and CBG enhanced immortalized cholangiocyte activity. The reduction in proliferation begins immediately and occurs maximally within 24 h of treatment. Moreover, a significant increase in the late-stage apoptosis and a reduction in the number of cells in S stage of the cell cycle indicates both CBD and CBG treatment could promote apoptosis and inhibit mitosis in CCA cells. The fluorescent expression of BAX and LC3b was significantly enhanced with CBD treatment when compared to control. LAMP1 and LC3b colocalization could also be observed with CBD and CBG treatment indicating changes in autophagic flux. A significant inhibition of migration, invasion and colony formation ability was shown in both CBD and CBG treatment in CCA. Western blot showed an overall decrease in the ratio of anti-apoptotic protein Bcl-2 with respect to pro-apoptotic protein BAX with CBG treatment. Furthermore, CBD treatment enhanced the expression of Type II cell death (autophagic degeneration) protein LC3b, which was reduced in CBG-treated CCA cells. Meanwhile, CBG treatment upregulated Type I cell death (programmed apoptosis) protein cleaved caspase-3. CBD and CBG are effective anti-cancer agents against CCA, capable of inhibiting the classic hallmarks of cancer, with a divergent mechanism of action (Type II or Type I respectively) in inducing these effects.

## 1. Introduction

Cholangiocarcinoma (CCA), a form of cancer arising from the bile duct, is on the rise worldwide, with incidence increasing 165% over the last 30 years from 0.32 per 100,000 to 0.85 per 100,000 [1]. Risk factors for CCA are highly variable and change by region, which presents challenges for assessing the cause of the increase in disease burden. CCA prognosis is usually quite poor due to no symptoms at the early stage and advanced tumor progression by the time of diagnosis in the late stage. The only effective treatment for CCA is surgery resection. However, the recurrence rates are about 49–64% after the surgical treatment [2]. Minimally successful chemotherapy strategies (median survival range of 4 to 10 months) are ultimately the treatment focus for many non-resectable tumors [3,4].

CCAs can be differentiated based on their origin within the biliary tree. They are either intrahepatic, within the liver; perihilar, at the junction of the left and right hepatic ducts; or distal, within the common bile duct [2]. The main cell line in this study, HuCC-T1, is an intrahepatic CCA sourced from a metastatic site in a 56-year-old male with accumulated mutations in MSH6, TP53 [5] and KRAS (which are associated with poor survival [6]). The other CCA line used, Mz-ChA-1, is a gall bladder carcinoma sourced from a metastatic site in a 55-year-old female with an accumulated mutation in ATM [7,8]. The normal cholangiocyte line, H69, is an SV40 transformed cell line with mutations in PIK3CA, RB1, and TP53 [9,10,11].

Presently, the most effective chemotherapy for CCA is a combination of gemcitabine and cisplatin (G/C) [12]. In HuCC-T1, gemcitabine induces cell cycle arrest in G1 without evidence of apoptosis [13,14]. Cisplatin induces DNA damage by cross-linking purine bases, thereby interfering with DNA repair processes [15]. This combination of cell cycle arrest and DNA damage leaves a lot to be desired in terms of patient survival and quality of life. Cannabidiol (CBD) and cannabigerol (CBG) represent readily available, non-psychoactive constituents of the cannabis plant. Many cannabinoids have shown promise in inhibiting breast cancer, prostate cancer, hepatocellular carcinoma, lung cancer, pancreatic cancer, lymphatic cancer and melanoma [16]. This is accomplished through stimulation of autophagy and activation of apoptosis or through autophagy-independent apoptosis pathways, chiefly through the accumulation of ceramides [17]. However, how CBD and CBG affects CCA is unknown. In this study, we aim to evaluate CBD and CBG as potential anti-tumor agents against CCA.

## 2. Materials and Methods

### 2.1. Materials

Reagents were purchased from Sigma-Aldrich (St. Louis, MO, USA), unless otherwise indicated. Rabbit polyclonal antibody BAX (Cat #5023), rabbit polyclonal antibody cleaved caspase-3 (Cat #9661), rabbit monoclonal antibody LAMP1 (Cat #9091), mouse polyclonal antibody LC3b (Cat #83506) and anti-rabbit linked HRP goat antibody (Cat #7074) were purchased from Cell Signaling Technologies (Danvers, MA, USA). Rat monoclonal antibody for Ki-67 (Cat #151202), rat monoclonal antibody GAPDH (Cat #607902), mouse monoclonal antibodies for β-actin (Cat #5714) and anti-mouse linked HRP goat polyclonal antibody (Cat #405306) were purchased from Biolegend (San Diego, CA, USA). Mouse monoclonal antibody for Bcl-2 (Cat #BCL2L2-1) was purchased from Developmental Studies Hybridoma Bank (Iowa City, IA, USA). Donkey anti-rabbit Alexafluor488 (Cat #ab150073), donkey anti-rat AlexaFluor555 (Cat #ab150154) and goat anti-mouse Alexafluor568 (Cat #ab175473) were purchased from Abcam (Cambridge, UK). CBD and CBG were obtained from Mile High Labs (Broomfield, CO, USA) and stored in filtered 20 mM DMSO stock solutions at −20 °C.

### 2.2. Cell Lines

HuCC-T1 (from intrahepatic bile ducts) cells were obtained from Dr. A.J. Demetris (University of Pittsburgh, Pittsburgh, PA, USA) and cultured as described [5]. Mz-ChA-1 cells are from a gallbladder, a gift from Dr. Alpini and originally from Dr. G. Fitz [8]. The human immortalized, non-malignant cholangiocyte cell line, H69 (from Dr. G.J. Gores, Mayo Clinic, Rochester, MN, USA) was cultured as described [18].

### 2.3. MTT Assay

Cell proliferation was measured in a 96-well assay (5000 cells per well) using MTT (3-(4,5-dimethylthiazol-2-yl)-2,5-diphenyltetrazolium bromide) 24 h after plating. Cells were treated with 0.1% dimethyl sulfoxide (DMSO) as a control or with CBG or CBD with serial dilutions ranging from 200 μM to 6.25 μM with 6 biological replicates per treatment. Cells were treated for 48 h, then assayed and measured with a spectrophotometer. To assay, MTT was added to the wells at 1:100 dilution MTT: Media, and allowed to incubate for 2 h, followed by aspiration of the media and addition of 8 mM ammonia suspended in DMSO. The plate was read at 540 nm after 10 min of incubation.

### 2.4. Cell Cycle analysis

Cells were plated in a 6-well plate (8 × 10^5^ per well), allowed to rest for 24 h, and then treated with either 0.1% DSMO, 100 μM CBD or 100 μM CBG for 24 h with 4 biological replicates per treatment. They were then stained with propidium iodide solution from Biolegend (San Diego, CA, USA, Cat #421301) according to manufacturer instructions. Cells were then measured for fluorescence using an Attune NxT Acoustic Focusing Cytometer from ThermoFisher Scientific (Waltham, MA, USA) and analyzed on the Attune NxT Software v3.1. 1.

### 2.5. Apoptosis assay

Cells were plated in a 6-well plate (8 × 10^5^ per well), allowed to rest for 24 h and then treated with either 0.1% DSMO, 100 μM CBD, or 100 μM CBG for 24 h with 4 biological replicates per treatment. They then were then stained with annexin V and propidium iodide using an Apoptosis Detection kit from Biolegend (San Diego, CA, USA, Cat #640914) according to manufacturer instructions. Cell fluorescence was then measured on an Attune NxT Acoustic Focusing Cytometer from ThermoFisher Scientific (Waltham, MA, USA) and analyzed on the Attune NxT Software v3.1.1.

### 2.6. Scratch Migration assay

Cells were in a 6-well plate (1 × 10^6^ per well) and allowed to grow to confluency over 48 h. Once 100% confluence was achieved, cells were scratched and then treated with either 0.1% DMSO, 100 μM CBD or 100 μM CBG in serum-free RPMI 1640 with 4 biological replicates per treatment. Distance of the resulting scratch was measured at 0, 6, 12, 18 and 24 h post treatment.

### 2.7. Immunofluorescence

Cells were plated in a 6-well plate (8 × 10^5^ per well) and allowed to rest for 12 h before being treated with either 0.1% DMSO, 100 μM CBD or 100 μM CBG with 4 biological replicates per treatment; cells were treated for 24 h for BAX immunostaining and 6 h for LC3b and LAMP1 co-staining. Cells were then washed with PBS, fixed with 10% neutral buffered formalin, permeabilized with ice-cold 100% methanol, blocked in 1% goat serum for 1 h and then stained with primary antibodies at a 1:500 dilution for 1 h at 37 degrees. The cells were washed and then stained with secondary antibodies at a 1:1000 dilution for 1 h before being counterstained with DAPI glycerol mounting solution (BAX was stained with Alexfluor555, LC3b was stained with Alexafluor488, and LAMP1 was stained with Alexafluor568).

### 2.8. Colonogenic Formation Assay

Cells were plated in a 6-well plate (2.5 × 10^5^ per well) and allowed to colonize for 7 days before being treated with either 0.1% DMSO, 100 μM CBD or 100 μM CBG for 7 days with 4 biological replicates per treatment. They were then washed with PBS, fixed in 10% NBF, and stained in a 0.5% crystal violet, 25% methanol solution. The size and number of the colonies were imaged and quantified with ImageJ software.

### 2.9. Transwell Migration Assay 

Cells were plated in a 24-well (2.5 × 10^5^ per well) transwell chamber (Greiner Bio-One, Monroe, NC, USA, Cat# 665638) according to manufacturer instructions. Cells were promptly treated with either 0.1% DMSO, 100 μM CBD, or 100 μM CBG for 16 h with 4 biological replicates per treatment. The upper chamber was then aspirated, washed with PBS, fixed in 4% neutral buffered formalin, and stained with 0.5% crystal violet, 10% methanol solution for 1 h before again being washed in PBS. Unmigrated cells were then removed from the upper chamber using a cotton swab. The size and number of the colony were imaged and quantified with ImageJ software.

### 2.10. Western Blot

Cells were plated in 100-mm dishes, grown to 90% confluency, and lysed. Protein content was measured with a Bradford assay (Cat# 23225) from ThermoFisher Scientific (Waltham, MA, USA). A total of 10 μg of protein per sample was resolved by SDS-PAGE and transferred to nitrocellulose membranes. Membranes were blocked with Superblock™ T20 (TBS) Blocking Buffer (Cat #37536) from ThermoFisher Scientific (Waltham, MA, USA) and incubated with the specific primary antibody at a 1:1000 dilution overnight at 4 °C, washed and incubated with secondary antibodies at a 1:5000 dilution for 1 h at room temperature (LC3b, Cleaved Caspase-3 and BAX were probed with anti-rabbit HRP, Bcl2 and β-actin were probed with anti-mouse HRP). Blots were stripped and re-probed with β-actin (*n* = 3 per group) or GAPDH (*n* = 4 per group) as a reference gene. Protein expression was visualized with an Azure 300 imaging system (Azure Biosystems, Dublin, CA, USA) and quantified with ImageJ software.

### 2.11. Colloidal Aggregation analysis

The size and polydispersity of CBD were determined using dynamic light scattering (DLS). The goal was to examine if the CBD forms colloidal aggregates. The cell culture media with and without DMSO which was used as solvent, CBD or CBG were analyzed for size and polydispersity using the DLS ZetaPALS Spectrometer (Brookhaven Instruments Corporation, New York, NY, USA). The viscosity of the media was set to 0.94 cP, whereas the refractive index was 1.345. The samples were run in triplicates, and each replicate sample was run three times. The average effective diameter and polydispersity were calculated for each sample.

### 2.12. Statistical Analysis

All data were analyzed using GraphPad Prism 9 software and reported as mean ± standard error mean (SEM) (San Diego, CA, USA). The Shapiro–Wilk test was performed to test for normality before parametric statistical tests were used. Following that, a one-way analysis of variance (ANOVA) was performed to test significant difference between group means for each experiment in this study. Tukey’s post hoc tests were then performed comparing every pair of groups to indicate which group is causing the significance in the ANOVA tests, when significance is detected. For all tests used, a *p*-value ≤ 0.05 was considered significant.

## 3. Results

### 3.1. Cannabinoids Reduced Proliferation and S Phase Cells in Cholangiocarcinoma

The cannabinoids CBD and CBG showed robust inhibitory activity in the HuCC-T1; MTT assay results showed a significant reduction in proliferation in the intrahepatic HuCC-T1 cells at the 100 & 200 μM concentration range for both CBD and CBG (Figure 1A). We confirmed this inhibitory activity as well in an extrahepatic bile duct CCA andMz-ChA-1 and found a significant reduction in proliferation of Mz-ChA-1 cells at 50, 100 and 200 μM concentrations of CBD and CBG (Appendix A). These anti-proliferative effects also occurred in immortalized cholangiocytes, H69, treated with CBD and CBG ranging from 50–200 μM (Appendix A). Conversely, a significant increase in proliferation was observed at the 25 μM dose for CBG in HUCC-T1 and from 6.25–12.5 μM in H69 cells (Appendix A). Since 100 μM of CBD and CBG treatment showed a significant inhibitory effect in CCA cells, it was then used for other functional assays. A time–course MTT assay at the 100 μM dose showed that significant reduction occurs sharply within the first 24 h, with maximal effect continuing to 48 and 72 h (Figure 1B). We found no interaction in anti-proliferative effects between the cannabinoids CBD and CBG when tested in conjunction with the standard chemotherapeutic intervention G/C (Appendix A). Furthermore, a significant reduction in the number of cells in the S phase occurred with both CBD and CBG treatment at 24 h post treatment (Figure 1C,D). There was also a significant reduction of cells in the S phase of CBG-treated cells when compared to CBD-treated cells (Figure 1C,D). We also quantified the expression of Ki-67 and measured the nucleus in immunofluorescence at 24 h post treatment using ImageJ software and noted the nuclei size was significantly reduced in both CBD- and CBG-treated cells. (Figure 1E,F and Appendix A). Although we did not find significant differences of the fluorescence density in both CBD and CBG treated groups, we did observe a trend of reduction in CBG-treated cells when compared to both control and CBD-treated CCA cells. We then checked for the presence of colloidal aggregates of CBG or CBG; the size and polydispersity of the drug particles were characterized using DLS (Appendix A). The observed effective diameter of the media increased from 63.4 nm to 70.0 nm following the addition of DMSO and stayed relatively the same with the addition of CBD or CBG. Overall, the effective diameter measured revealed that the CBD or CBG forms no significant colloidal aggregates. The polydispersity data range from 0.35–0.37, suggesting homogeneity of the particles.

### 3.2. Cannabinoids Prevented Wound Closure and Attenuated Migration

In the scratch migration assay, both CBD and CBG treatment significantly reduced the rate of wound closure at the 18-h and 24-h time points (Figure 2A,C). A significant reduction of migrated cells was shown in both CBD and CBG treated groups, with even more reduction occurring with CBG treatment when compared to CBD (Figure 2B,D). In the colonogenic formation assay, a significant reduction in the number and size of colonies was observed only in CBG treatment (Figure 2E–G).

### 3.3. Cannabidiol and Cannabigerol Induce Cell Death by Different Mechanisms, through Autophagic Pathways and Apoptotic Pathways, Respectively

When analyzing cell apoptosis using annexin V propidium iodide co-staining, we found a significant increase in the number of cells in the late stage of apoptosis in both CBD and CBG treatments, with a significant increase occurring in CBG treatment when compared to CBD (Figure 3A,B). A significant increase in fluorescence intensity of BAX, a pro-apoptotic protein, was observed with CBD treatment but not in CBG treatment (Figure 3C). This result is reflected in the Western blot analysis of BAX expression (Figure 4A,B). Although the expression of BAX is not significant, the trend is consistent with the immunofluorescence. Furthermore, an increase in the ratio between BAX (pro-apoptotic) and Bcl-2 (anti-apoptotic) was observed in both CBD and CBG treatment groups, primarily due to a decrease in the expression of Bcl-2 (Figure 4A,B). The Type 1 cell-death pathway, noted by cleaved caspase-3, showed an increase in CBG treatment (Figure 4C,D); with the Type 2 cell death pathway, noted by LC3b, showing a clear significant increase in CBD and significant decrease in CBG treatment compared to CBD (Figure 4E,F). We then assessed autophagic flux through co-staining LC3b and LAMP1 and found a significant increase in fluorescence intensity of LC3b with CBD treatment and an observed increased with CBG treatment that was not significant due to low end deviation within CBG treatment (Figure 5A,B). It is interesting to note that the LAMP1 expression was not changed, but the colocalization of LC3b with LAMP1 occurred in both CBD and CBG treatment.

## 4. Discussion

In this study, we found that both CBD and CBG were effective in inhibiting cholangiocarcinoma cells in vitro in a dose-dependent manner, with CBG being significantly more effective at the same dose as CBD across several functional assays: MTT, wound closure, transwell, colony formation and apoptosis. We found that CBD and CBG induce their cytotoxic effects via different mechanisms, with CBD stimulating autophagic and apoptotic pathways and CBG stimulating only apoptotic pathways.

Initially, we examined the cytotoxicity of CBD and CBG by establishing a dose-response curve, as well as a time-response curve, for the CCA cell line. We found that effects occurred immediately and within 24 h. We confirmed that these effects were not specific to just the intrahepatic cell line. By testing another CCA of different origin, Mz-ChA-1 and found that the cannabinoids were still effective. Interestingly, we found that the immortalized cholangiocytes were more sensitive to the cannabinoids, with cytotoxicity occurring at one-fourth the dose in CCA. This might suggest that the receptor responsible for mediating this mechanism occurs at a higher density in the immortalized cholangiocytes. It also suggests that these cannabinoids could induce toxicity to non-malignant cholangiocytes, consistent with the potential hepatotoxicity of the current FDA-approved version of CBD, epiodolex, that limits its dosage. The difference between the activities of the two cannabinoids has also been observed in crude hemp extract in H69 cells [19] where CBG appears to be more active than CBD. Thus, this potential toxicity should be carefully considered in clinical situations. However, it is interesting to note that H69 and HuCC-T1 have similar mutation in TP53, which may contribute to the toxicity observed in this model [5,10]. We further tested if CBD and CBG caused non-specific toxicity due to the presence of colloidal aggregates in solution, a phenomenon that has been suggested as a mechanism for cell death in a number of nutraceutical therapeutics [20] and which has been observed in CBD [21]. Our experiment failed to show an aggregate within the cell media. We also observed increases in the proliferation of HuCC-T1 cells at lower doses of CBD and CBG, the biphasic dose response termed “hormesis” that is observed in chemotherapeutics where it is defined by low dose stimulation and high dose inhibition [22].

We chose to use the 100 μM dose for CBD and CBG in downstream assays due to the significant and robust response to both drugs at this dosage. It has been suggested that high doses of cannabinoids can induce non-specific cytotoxicity by forming colloidal aggregates due to their low solubility [21]; however, we failed to find evidence of this phenomena within our experimental parameters. Evaluating the drugs for any influence on cell cycle, we found that both cannabinoids induced cell cycle arrest at G0/G1 and caused a significant reduction in the number of cells in the S phase. This effect has previously been observed with CBD in gastric cancer cells [23]. Cell cycle arrest can be caused by various external stresses, and with pleiotropic activity within the cell, cannabinoids could also cause cell cycle arrest through various observed mechanisms. It has been previously reported that within this close range in HepG2 cells, CBD induces DNA damage, which is a known and plausible cause of cell cycle arrest [24]. Additionally, CBD pharmacologically interacts with transient receptor potential channels [25], a phenomena known to induce cell cycle arrest from capsaicin [26]. Transient receptor potential channel activation has been suggested as a potential strategy in treating cancer and overcoming drug resistance [27].

Next, we performed immunofluorescent staining of the nucleus and proliferative marker Ki-67, an abundant protein in the nucleus of tumor cells during active stages of the cell cycle G1,G2,S and M but not G0 [28]. Qualitatively, our results showed a loss of Ki-67 foci with CBG treatment that was present in the control, as well as associated Ki-67 phenotype with cells exiting active phases of the cell cycle and entering G0, which is what we observed earlier in our propidium iodide analysis. Additionally, we observed significant nuclear condensation with cannabinoid treatment, another peculiar phenotype associated with senescence and apoptosis [29,30]. Nuclear condensation occurs in three distinct stages: (1) ring condensation, marked by underlying structural changes but no change in nuclear size; (2) necklace condensation, marked by a bulk movement of chromatin inwards; and (3) nuclear collapse disassembly, marked by fragmentation of the nucleus and DNA to form apoptotic bodies [31]. When evaluating our own micrographs, CBD treatment appears to be in Stage 2, with reduction of size of the chromatin, while CBG appears to have completed collapse and disassembly. This is another indication of the differences in mechanistic action between the two cannabinoids.

We then evaluated the effects of CBD and CBG on the metastatic qualities of CCA using HuCC-T1 (HuCC-T1 originated from a metastatic site). Both cannabinoids showed a strong ability in reducing the rate of wound closure, transwell migration, and colony formation, with CBG being significantly more effective than CBD at inhibiting transwell migration. In another migration-based assay, similar inhibition was observed in glioblastoma, cells with CBG again appearing to induce a stronger response [32]. Colony phenotype also changes during the colonogenic formation assay, with the size of CBD colonies increasing despite a decrease in the number and CBG showing both a decrease in the size and number of colonies.

Beyond a significant change in the transwell assay, CBG also presented significantly more cells in late-stage apoptosis than CBD when assayed with a cell-death flow cytometry kit where both drugs increased in apoptotic cells. This supports many of the observations made previously in this study comparing the two drugs. CBG has shown to be more effective than CBD in reducing ultraviolet-light-induced inflammatory cytokines, such as tumor necrosis factor-α and interleukin-6 [25]. CBG is more effective in reducing the *Cutibacterium acne-* induced interleukin 1-beta production in primary normal human epidermal keratinocytes [33]. Immunofluorescent staining of BAX revealed heavy expression of the pro-apoptotic mitochondrial pore protein from CBD treatment but not CBG; again, the DAPI staining show similar nuclear condensation patterns to the Ki-67 staining performed. The induction of the mitochondrial apoptotic pathway by CBD has also been observed in gastric cancer cells [23], as well as acute lymphoblastic leukemia, where dysfunction was driven by the release of intracellular calcium stores [34]. When we further evaluated BAX through Western blot, we also measured Bcl-2 levels; Bcl-2 is an anti-apoptotic protein responsible for inhibiting BAX. Though we found no quantitative difference in the expression of BAX, we found a reduced level of Bcl-2, suggesting a shift toward pro-apoptotic cellular environment.

In light of this, we evaluated cleaved caspase-3 expression as well as LC3b. Cleaved caspase-3 is an executioner caspase responsible for mediating DNA fragmentation and nuclear collapse leading to apoptosis, and LC3b is an essential component to autophagic processes within the cell [35,36]. Here we found that CBD and CBG had elevated expression of cleaved caspase-3, with CBG showing a slight lead. There was a clear distinction between CBD and CBG expression of LC3b, with CBD showing a clear and significant elevation of the autophagic protein, and CBG inverting that and showing a significant decrease. We then assessed the autophagic flux of LC3b and a combination with LAMP1 which showed a significant increase in LC3b fluorescence with CBD and a trend of increase in CBG treatment, as well as further colocalization with LAMP1 protein. LAMP1 is a common lysosomal marker, and colocalization of LC3b along with LAMP1 confirms formation of an integral autophagic process, the autophagosome, occurring with cannabinoid treatment [37]. The increase in autophagic processes has also been observed in a study examining CBD as an intervention in a murine alcoholic fatty liver disease model [38]. Taking all of this into account, what we see is that the cellular signaling method from CBD involves mitochondrial-dependent caspase activation and autophagic upregulation, while CBG activity is mitochondrial independent.

As shown in Figure 6, we suggested transient receptor potential channels as a potential mechanism of action driving this process. Previous comparative pharmacological evaluation of CBD and CBG at these receptors found that CBD agonizes transient receptor potential channels a level of magnitude more than CBG, yet both desensitize these channels equally, which could explain the differences in response [39]. CBD and CBG also diverge pharmacologically at other intracellular receptors: CBD is a known agonist of PPARγ [40], while CBG exerts strong activity as an alpha-2 adrenergic agonist [41]. PPARγ has been found to be essential to CBD-induced apoptosis in lung cancer in vitro [42], and it has also been implicated in upregulating autophagy [43]. Alpha 2 adrenergic activation has been linked to caspase-dependent apoptosis in several studies [44,45,46]. Overall, though, this receptor pathway is unproven but should warrant future investigation to further elucidate the specific therapeutic target of this therapy.

In summary, our study showed the anti-tumor effect of CBD and CBG on CCA growth in vitro. CCA seems to be more sensitive to the CBG treatment compared to CBD. We have successfully demonstrated that CBD and CBG could induce cell death and inhibit migration and invasion in CCA cell lines. Therefore, CBD and CBG could be potential cannabinoids to treat CCA. However, the dosage used in our study also showed cytotoxicity in immortalized cholangiocytes. Finally, the study found that CBD and CBG inhibit tumor growth via different cell death pathways.

## Figures and Tables

**Figure 1 biomolecules-12-00854-f001:**
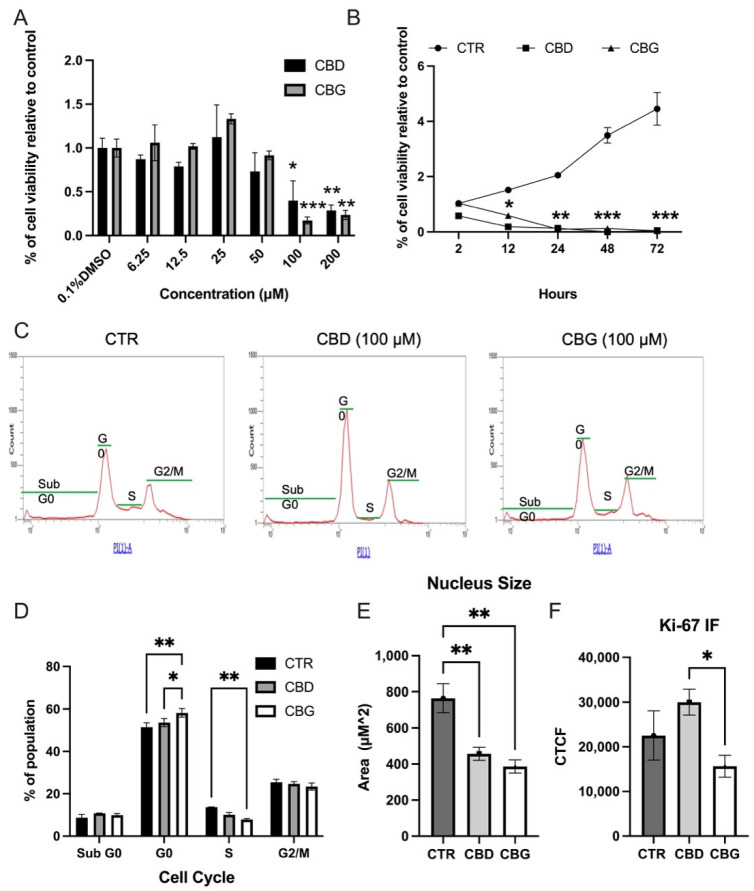
CBD and CBG treatment reduced proliferation in HuCC-T1 cells: (**A**). MTT assay results showing cell viability in HuCC-T1 48-h post treatment with control (0.1% DMSO), CBD or CBG; (**B**). time–course MTT assay at 100 μM established dose over a 72-h time period; (**C**). propidium iodide histogram showing cell cycle doublet shown; population in each group (Sub G0, G0,S,G2/M) shown; (**D**). percentage of each cell cycle population in each treatment was quantified, *n* = 3; (**E**,**F**). quantification of Alex Fluor 488 staining of Ki-67 (green) and DAPI staining of nuclei size; (**E**). and fluorescence intensity; (**F**). in all three groups with a significant decrease in the size of the nucleus; (**F**). CTCF: corrected total cell fluorescence. * *p* < 0.05 vs. 0.1% DMSO or CTR; ** *p* < 0.01 vs. 0.1% DMSO or CTR; *** *p*< 0.001 vs. 0.1% DMSO or CTR.

**Figure 2 biomolecules-12-00854-f002:**
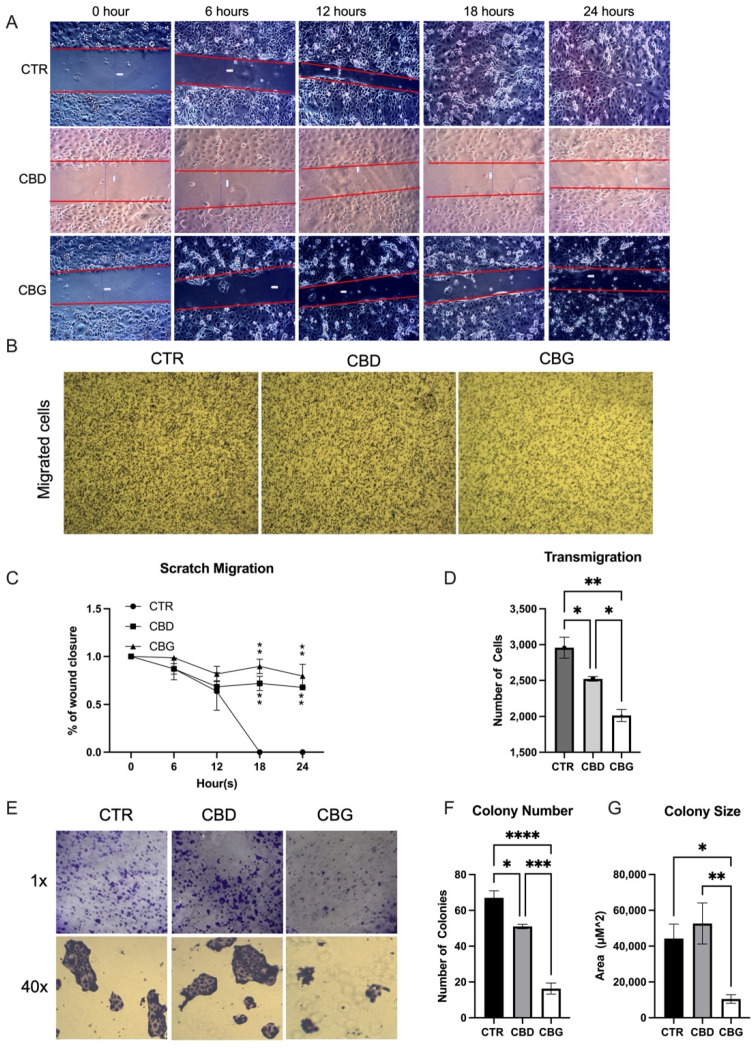
CBG treatment inhibited the wound healing and migration: (**A**). scratch migration assay, the diameter of the wound was measured every 6 h over a 24-h period; (**B**). micrographs showing cells plated in a Transwell 48 h after seeding and treatment with either 0.1% DMSO (CTR), 100 μM CBD or 100 μM CBG; (**C**). percentage of wound closure relative to the initial size of the wound; significant difference at 18 h and 24 h in treatments (CBG and CBG) vs. control; (**D**). quantification of colonies using ImageJ shows a significant decrease in cell migration number in CBD treatments and a significant decrease from CBD to CBG treatment; (**E**). colonogenic formation assay showing cells 7 days after being plated as single cell colonies and then treated with CTR, CBD or CBG (100 μM) for 24 h; (**F**,**G**). quantification of colonies shows a decrease in colony formation number in both CBD and CBG treatments and a significant decrease in colony size with CBG treatment. * *p* < 0.05 vs. CTR, ** *p* < 0.01 vs. CTR, *** *p* < 0.001 vs. CTR, **** *p* < 0.0001 vs. CTR.

**Figure 3 biomolecules-12-00854-f003:**
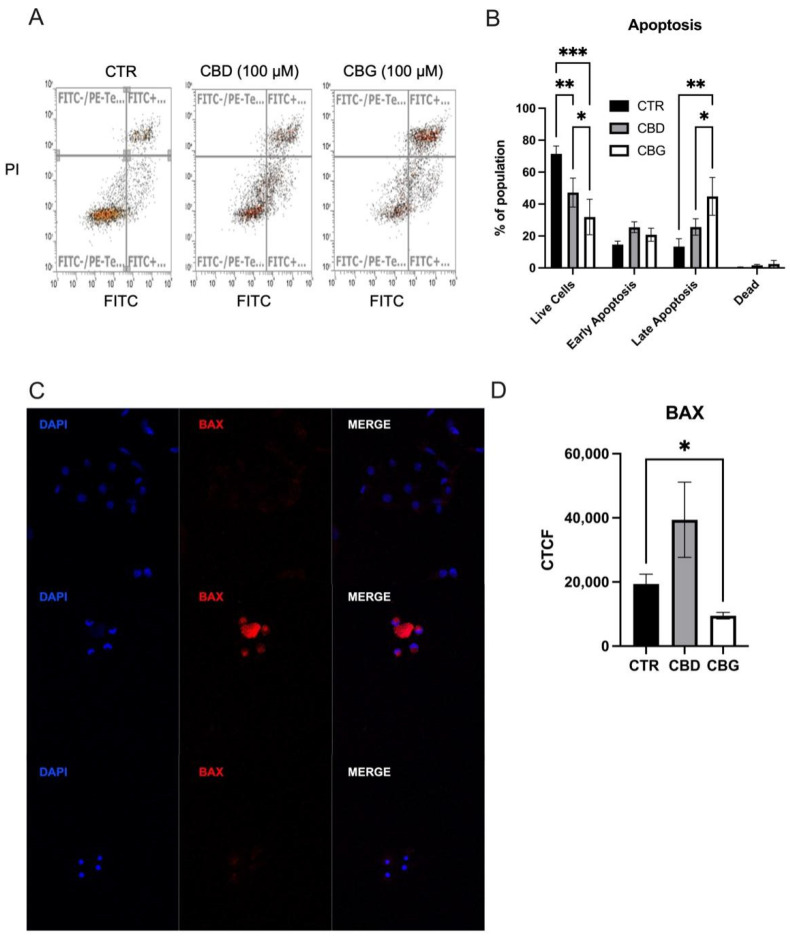
CBD and CBG treatment promoted apoptosis and autophagy in HuCC-T1 cells; (**A**,**B**). annexin V (FITC-H) and propidium iodide (PI) co-staining flow cytometry analysis showing an increase in late apoptotic cells and a significant decrease in live cells upon CBG treatment; (**C**). DAPI staining of nucleus (blue) and Alex Fluor 555 staining of BAX (red) for 0.1% DMSO (top), 100 μM CBD (middle) and 100 μM CBG (bottom) at 24 h post treatment; (**D**). graph showing quantification of BAX fluorescence in all three groups with a significant change in BAX fluorescence with CBG treatment. CTCF: corrected total cell fluorescence. * *p* < 0.05, ** *p* < 0.01, *** *p* < 0.001.

**Figure 4 biomolecules-12-00854-f004:**
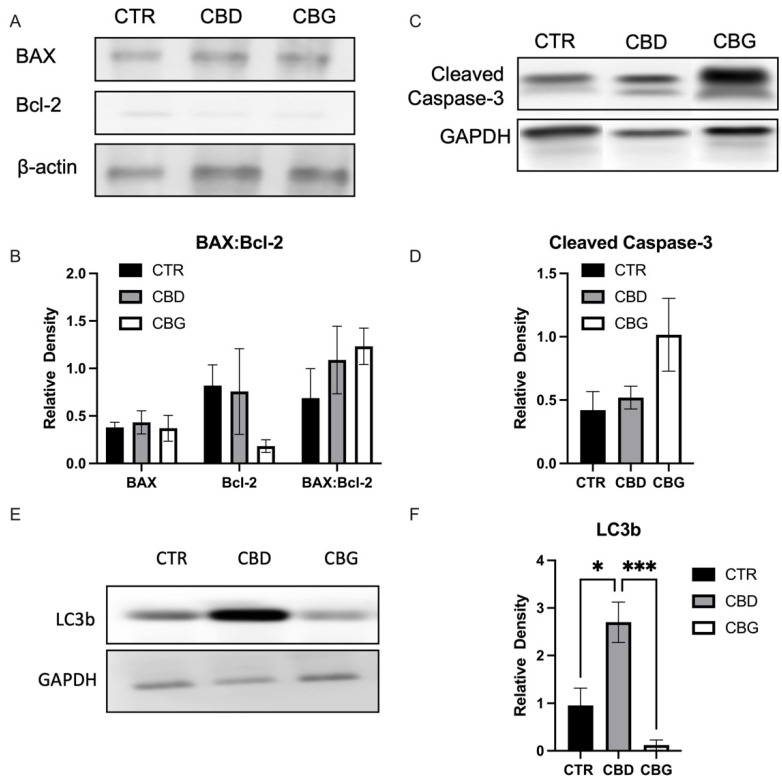
CBD and CBG induces apoptosis in a divergent cell-death pathway in HuCC-T1 cells; (**A**). representative image of Western blotting of BAX (pro-apoptotic) and Bcl-2 (anti-apoptotic) protein and β-actin (reference); (**B**). quantification of the BAX, Bcl-2 and ratio of BAX to Bcl-2; (**C**). representative image of Western blotting of cleaved caspase-3 (Type 1 cell death, apoptosis) and GAPDH (reference) shows trend toward increase in CBG treatment; (**D**). quantification of cleaved caspase-3; (**E**). representative image of Western blotting of total LC3b (Type 2 cell death, autophagy) (**F**). Quantification of total LC3b. * *p* < 0.05 *** *p* < 0.001.

**Figure 5 biomolecules-12-00854-f005:**
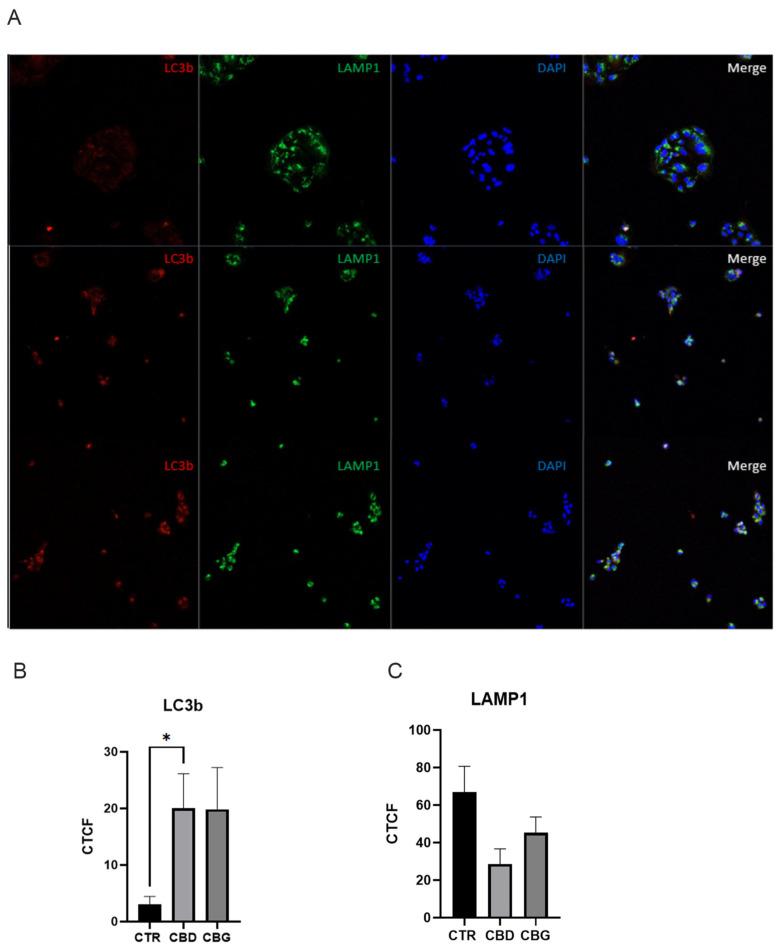
CBD and CBG treatment promoted autophagy in HuCC-T1 cells; (**A**). Alexfluor568 staining of LC3b (red) Alexfluor488 staining of LAMP1 (green), and DAPI staining (blue) of the nucleus for 0.1% DMSO (top), 100 μM CBD (middle) and 100 μM CBG (bottom) at 6 h post treatment; (**B**). graph showing quantification of LC3b fluorescence in all three groups with a significant increase in LC3b fluorescence with CBD treatment and an increase with CBG treatment (*p* = 0.108); (**C**). graph showing quantification of LAMP1 fluorescence in all three groups with no significant changes between groups. CTCF: corrected total cell fluorescence. * *p* < 0.05.

**Figure 6 biomolecules-12-00854-f006:**
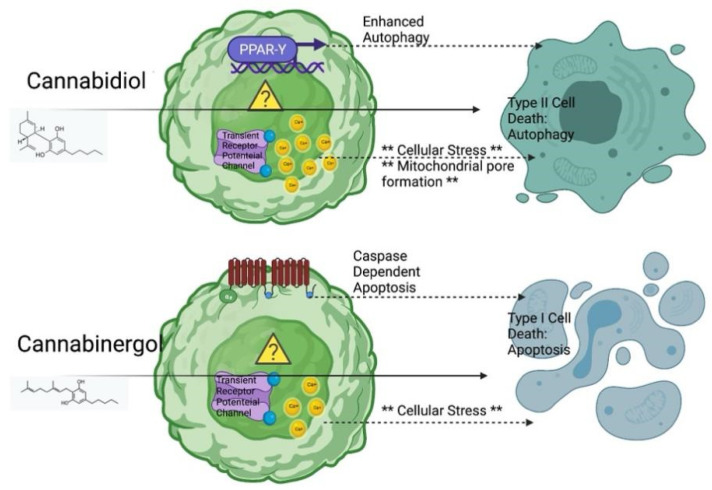
Proposed Mechanism of cannabidiol and cannabigerol against cholangiocarcinoma. Figure is generated from BioRender.com by authors (accessed on 6 June 2021).

## Data Availability

The data presented in this study are available on request from the corresponding author.

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
