# Peer review of "Cannabidiol and Cannabigerol Inhibit Cholangiocarcinoma Growth In Vitro via Divergent Cell Death Pathways"

_biomolecules, 2022, doi:10.3390/biom12060854_

Round 1
Reviewer 1 Report
I appreciated the authors' point-by-point responses to my comments.
In my opinion the revised version of the manuscript is suitable for publication.
Congratulations
Reviewer 2 Report
The Authors satisfactority answered the questions raised by the reviewer. The manuscript is improved and worthy to be accepted
This manuscript is a resubmission of an earlier submission. The following is a list of the peer review reports and author responses from that submission.
Round 1
Reviewer 1 Report
The manuscript is well written; however, very important details were not reported in the materials and methods section and some results were treated and discussed superficially. Therefore, in my opinion, the manuscript needs major revision.
The main criticism
The antiproliferative effect of high doses of CBD and CBG may be due to the fact that these cannabinoids were used beyond their limit of solubility in an aqueous solution, as the culture medium.
It is known that cannabinoids are hydrophobic molecules poorly soluble in aqueous solution, and several studies are being aimed at developing nanoformulations to improve their bioavailability.
It is very likely that CBD when administered at a concentration of 100µM may precipitate and give rise to toxic colloidal aggregates. In fact, Dynamic Light Scattering studies have shown that CDB forms colloidal aggregates in the culture media at much lower concentrations than those used by the authors (Nelson KM et al 2020 The Essential Medicinal Chemistry of Cannabidiol (CBD). J Med Chem. 63 (21): 12137-12155. Doi: 10.1021 / acs.jmedchem.0c00724.). It has also been shown that these aggregates perturb the integrity of the plasma membrane and therefore could be the cause of the cytotoxicity shown on cholangiocarcinoma cells treated with 100µM CBD.
For this reason, the authors must evaluate the state of aggregation of the cannabinoids used in their experiments, with spectroscopic or other methods. The aggregation state could explain the differential effect of cannabinoids when supplied at low concentrations (pro-proliferative! Fig1A and Suppl Fig2) and high concentrations (anti-proliferative!).
Other critical issues:
It is not indicated the source of cannabinoids used in the study, nor how their stocks were generated and stored.
Methodological details are lacking on how the MTT assay was performed, and which instrument and software were used for FACS analysis.
The authors did not indicate how many biological replicates and technical replicates were performed for the experiments and these data are fundamental for interpreting the statistical significance of the results!
It is known that the scratch migration assay must be performed in conditions of serum limitation or starvation or in the presence of antimitotic agents to prevent cell proliferation from interfering with the wound closure due to migration. The authors should specify the culture condition of the scratch migration assay and better discuss the results.
The dilutions of the primary antibodies and if they were diluted in milk or BSA, and the details of the secondary antibodies used should be reported.
The antibody catalog numbers do not match those reported on the Cell Signaling Technology site, for example for Antibody LC3b → cat # 803506T. The authors should indicate which band of LC3b was shown in the western blot result reported in Fig4 to speculate about autophagy. In this case, it would be important to see the whole Wb to highlight the bands corresponding to LC3-I and LC3-II.
Furthermore, autophagy is not a cell death process. Indeed, it is a survival process that allows the cell to protect itself from the presence of toxic aggregates and dysfunctional organelles, etc…
The authors should rewrite the results considering this important aspect of autophagy and better discuss the implications of CBD's promotion of autophagy.
It is very important that the authors, before supporting the use of cannabinoids in the treatment of cancer patients, discriminate between their real anticancer effects and non-specific toxicity!
Reviewer 2 Report
The work of Viereckl and colleagues is focused on studying the effects of two components of cannabis, cannabidiol (CBD) and cannabigerol (CBG), on the biology of cholangiocarcinoma (CCA). CCA is an extremely aggressive tumor with no effective pharmacological therapies whose biology is still poorly understood. The main results obtained by Authors on in vitro cultures of HUCCT-1 cells derived from intrahepatic CCA are: a) both CBG and CBD are able to inhibit cell proliferation and cell viability; b) The two cannabinoids decrease its motility and the ability to form colonies; c) CBD is more involved in autophagy mechanisms, while CBG is more involved in apoptotic mechanisms. The work is well written and data clearly exposed. Data are promising and certainly of potential interest, but despite this, some experimental flows dampen the enthusiasm for this work.
The main problem with this work is that it is based on a single cell line of intrahepatic origin and, given the genetic variability of CCAs, these results could be cell-dependent. It would be necessary to reproduce at least the main experiments I other CCA cell lines with different genetic background.
The second problem is that the pictures are grainy, especially the immunofluorescences. These should be exchanged for better quality photographs.
Figure 1A: to better evaluate the degree of cell viability in relation to the “real life” of CCA treatments, the reviewer suggests adding the treatment with cisplatin + gemcitabine, which is currently the gold standard for the treatment of advanced CCA.
To better assess whether it is early or advanced apoptosis, I suggest doing a double immunoluorescence LC3b + LAMP1 in cells with and without drug treatment.